# The use of a patch to augment rotator cuff surgery – A survey of UK shoulder and elbow surgeons

**M. J. Baldwin**[1]*, **N. S. Nagra**[1], **N. Merritt**[1], **J. L. Rees**[1], **A. J. Carr**[1], **A. Rangan**[1,2], **M. Thomas**[1], **D. J. Beard**[1], **C. Cooper**[1], **L. Kottam**[2], **J. A. Cook**[1]

**1** Nuffield Department of Orthopaedics, Rheumatology and Musculoskeletal Sciences (NDORMS), Botnar Institute of Musculoskeletal Sciences, University of Oxford, Oxford, United Kingdom, **2** The James Cook University Hospital, South Tees Hospitals NHS Foundation Trust, Middlesbrough, United Kingdom

* matthew.baldwin@ndorms.ox.ac.uk

**Data Availability Statement:** Anonymised dataset provided as supplementary S2 File.

**Funding:** We would like to thank all the BESS members who kindly completed this survey.

## Abstract

### Background

Rotator cuff tears are a common cause of shoulder pain and can result in prolonged periods of pain, disability and absence from work. Rotator cuff repair surgery is increasingly used in an attempt to resolve symptoms but has failure rates of around 40%. There is a pressing need to improve the outcome of rotator cuff repairs. Patch augmentation increasingly being used within the NHS in an attempt to reduce repair failures. The aim of this survey was to determine current UK practice and opinion relating to the factors that influence choice of patch, current patient selection and willingness to assist with generation of improved evidence.

### Methods

An online survey was sent to the surgeon members of the British Elbow and Shoulder Society (BESS). Questions covered respondent demographics, experience with patches, indications for patch augmentation and willingness to be involved in a randomised trial of patch augmented rotator cuff surgery.

### Results

The response rate was 105/550 (19%). 58% of respondents had used a patch to augment rotator cuff surgery. 70% of patch users had undertaken an augmented repair within the last 6 months. A wide surgical experience in augmentation was reported (ranging 1 to 200 implants used). However, most surgeons reported low volume usage, with a median of 5 rotator cuff augmentation procedures performed. At least 10 different products had been used. Most of the patches used were constructed from human decellularised dermis tissue, although porcine derived and synthetic based patches had also been used. Only 3–5% stated they would undertake an augmented repair for small tears across ages, whereas 28–40% and 19–59% would do so for large or massive tears respectively. When assessing patient suitability, patient age seemed relevant only for those with large and massive tears.

PARCS is funded by the NIHR Health Technology Assessment Programme (15/103/03). JAC is partly funded by the NIHR Oxford Biomedical Research Centre (NIHR-BRC- 10 1215-20008). The research was also supported by the National Institute for Health Research (NIHR) Oxford Biomedical Research Centre. The views expressed are those of the author(s) and not necessarily those of the NHS, the NIHR or the Department of Health. The funding body had no role in the design of the study and the collection, analysis and interpretation of data; in writing the manuscript; or in the decision to submit manuscripts for publication.

**Competing interests:** Author AC has been developing a patch for potential use to rotator cuff surgery. None of the other authors have any COIs to declare.

Half of the surgeons reported an interest in taking part in a randomised controlled trial (RCT) evaluating the role of patch augmentation for rotator cuff surgery, with a further 22% of respondent's undecided.

## Conclusions

A variety of patches have been used by surgeons to augment rotator cuff repair with a wide range of operator experience. There was substantial uncertainty about which patch to use and differing views on which patients were most suitable. There is a clear need for robust clinical evaluation and further research in this area.

## Background

Rotator cuff tears are a debilitating condition that results in pain, weakness, reduced shoulder mobility and sleep disturbance. It is estimated that the overall prevalence of full-thickness tears is between 15–20% with the rate set to increase as populations age.[1,2] While a large proportion are asymptomatic, many symptomatic full-thickness tears will often require surgical repair. Indeed, 9,000 rotator cuff repairs are performed each year in the NHS in England alone, at a cost of £6,500 per operation.[3] Unfortunately, in spite of surgical repair, there is a current failure rate of tendon healing of up to 40% with greater patient age and tear size both predictive of failure.[4,5] While various surgical techniques have attempted to improve the outcome of rotator cuff repair, there remains a real need to improve healing rates.

One promising approach is the use of a patch to augment the repair. Patches can be broadly grouped into two categories based on the materials used; biological and synthetic patches. The removal of cellular components from xeno- or allografts to produce a decellularised extracellular matrix (ECM), is a common method of biological scaffold production. Alternatively, synthetic patches can be produced from a variety of polymers. The use of Teflon (polytetrafluoroethylene) grafts in the repair of massive cuff tears, was first reported over 30 years ago.[6] More recently, the number of patches developed specifically for use in rotator cuff surgery has increased significantly. However, there is currently limited evidence to inform clinical decisions on when to utilise patches and which type is most appropriate. Of the two RCTs identified in a previous review, one evaluated a patch derived from porcine small intestine that has since been removed from the market due to safety concerns.[7,8] It is unclear what the current practice is regarding patch use.

The aim of this survey was to identify current UK clinical practice and gather information on surgeon opinion relating to the factors that influence their choice of patch, and patient suitability for patch augmented rotator cuff repair. Additionally, we aimed to ascertain the degree of support amongst shoulder surgeons for a randomised trial into patch augmentation.

## Methods

### Administration of survey

Surgical members of the British Elbow and Shoulder Society (BESS) were invited via email to participate in an online survey prepared using the Bristol Online Survey system. The main mechanism of approaching participants was via the BESS email list; the invite email was sent out by the BESS office to avoid unnecessary sharing of personal data. Information about the PARCS (Patch Augmented Rotator Cuff Surgery) feasibility study[9] and a hyperlink to the survey was provided. The survey was designed to take approximately 10 minutes to complete.

There was no minimum number of responses required as the study was opportunistic in terms of sample size and not driven by statistical testing. The response rate was defined as the number of responding participants divided by the number of eligible people invited. The statistical analysis was descriptive only. Responses were summarised quantitatively or narratively, as appropriate (using Microsoft Excel (Version 16.12) and Prism (Version 7.0)). No attempt was made to validate individual responses.

The email invitation was sent out on the 6th April 2017 and the survey closed on the 31st August 2017. Surgeon members of BESS attending the 2017 annual meeting were also offered an opportunity to complete the survey at the meeting using a laptop at an exhibition stand if they had not already completed it. This was a voluntary survey of health care professionals therefore formal ethical review was not sought. However, the survey was approved by the British Elbow and Shoulder Society (BESS) committee. A formal consenting process was not undertaken, rather completion of the survey was taken as implied consent.

## Survey contents

To assess respondent demographics participants were asked about their grade (consultant, trainee, other) and place of work (district general hospital, university teaching hospital, private practice, other). To determine the degree of exposure to augmented rotator cuff repair we asked about their preferred surgical technique for rotator cuff repair (predominantly open, predominantly arthroscopic, substantial amount of both open and arthroscopic repairs), whether they had previously used patch augmented rotator cuff surgery (no, yes within 6 months, yes but not within 6 months) and the total number of augmented cuff repairs that they had undertaken.

The next section sought to determine the types of patch commonly in use and investigate the factors influencing patch selection. Two separate free text questions were posed; which patches have you used? and why did you use these specific patches? A final free text space was provided to allow further comments relating to the choice of patch to be recorded.

To address the controversies surrounding patient selection for augmented rotator cuff repair we asked respondents to consider discrete patient subgroups. Age and tear size have been suggested to be the two main factors affecting outcome after rotator cuff surgery. 16 combinations of age and tear size were provided and, for each, asked respondents if they considered patch augmentation appropriate. Four different tear sizes (small, medium, large and massive) were combined with different four different ages (50, 60, 70 and 80 years old) to produce 16 combinations. An answer of 'Yes', 'No' or 'Unsure' could be provided for each scenario. A free text space was then provided to capture further comments relating to patient suitability.

Respondents were then asked to consider participation in a future clinical trial into augmented rotator cuff repair. Responders were asked whether they would be interested in participating in a randomised control trial (RCT) of patch augmented surgery (yes, no, maybe). Finally, we asked members what factors could be addressed to encouraged participations in a RCT.

Within our department we piloted our survey on four members of the shoulder and elbow surgical team who performed patch repair.

A copy of the survey can be found in S1 Survey.

## Results

### Respondents characteristics

All 550 medically qualified members of BESS were invited to participate with 105 (19%) responding. The respondents were mostly consultant surgeons (97%) with the majority

working at district general hospitals (48%) (Table 1). Most participants (95%) worked within the National Health Service (NHS) but with 32% reporting additional work within the private sector.

## Experience with rotator cuff augmentation

Most respondents undertook arthroscopic rotator cuff repairs (66%) with only 14% solely undertaking open repairs (Table 2). When asked whether they had ever used a patch to augment rotator cuff surgery over half (58%) had done so, with 70% of patch users undertaking an augmented repair within the last 6 months. Interestingly, the utilisation of patches among surgeons performing open repairs was slightly lower (40%) than for those reporting an arthroscopic (56%) or mixed open and arthroscopic practice (76%).

A varied surgical experience in augmentation was reported amongst those who had carried out a patch augmented rotator cuff repair, ranging from 1 to 200 procedures. However, most surgeons reported low volume usage, with a median of 5 rotator cuff augmentation procedures performed (Table 2).

## Patch type

When asked about the patches utilised during rotator cuff repair, 13 different products were reported (Table 3). Decellularised dermis accounted for 85% of the different patches utilised and non-degradable synthetic meshes the remaining 15%. Human decellularised products were more frequently used, with only 11% of decellularised patches being porcine derived and the rest being human. All reported synthetic scaffolds were non-degradable and produced from a variety of polymers (Polyester, Polypropylene and Polyurethane). Overall, Graft Jacket® was the most commonly reported device (55%) with the Leeds-Kuff Patch™ (10%), Arthroflex® (8%) and dCell® (8%) the next most common.

Reported factors influencing patch selection are given in Table 4. The device's perceived efficacy was an important theme, with clinical evidence (24%), personal (8%) and peer experience (4%) cited as important determinants. Product characteristics formed another dominant theme, with a patches' usability, strength and material influencing selection. Interestingly, a product's cost as well as availability within the local hospital, were also important. Finally, the specific characteristics of a rotator cuff tear may also determine patch choice.

## Patch indications

Responses for patients aged 50 or 60 years, and 70 or 80 years tended to be similar (Fig 1) regarding patient suitability for receiving a patch. However, the effect of age on patient

**Table 1. Training grade and place of work of respondents.**

| Category | n | n (%) |
|---|---|---|
| *Training Grade* | 105 | |
| Consultant | | 102(97) |
| Associate Specialist | | 1(1) |
| Orthopaedic Trainee | | 2(2) |
| *Place of Work* | 105 | |
| District General Hospital (DGH) | | 50(47) |
| Teaching Hospital | | 44(42) |
| Mixed—DGH + Teaching Hospitals | | 6(6) |
| Private Hospital | | 5(5) |

**Table 2. Surgeon reported experience with patch augmentation.**

| Category | n* | n (%) |
|---|---|---|
| *Preferred repair technique* | 105 | |
| Open | | 69(66) |
| Arthroscopic | | 15(14) |
| Open or Arthroscopic | | 21(20) |
| *Use of patch augmentation* | 105 | |
| Yes–within 6 months | | 43(41) |
| Yes–not within 6 months | | 18(17) |
| No | | 44(42) |
| *Number of patches implanted#* | 61 | |
| 1–5 | | 32(30) |
| 6–10 | | 15(14) |
| 11–15 | | 0(0) |
| 16–20 | | 6(6) |
| >20 | | 8(8) |

*n refers to the number of respondents.

#Total number of patches refers to the total number of patches used during each respondents surgical career to date.

suitability was clearly influenced by tear size. Among older patients (70 or 80 years) with small and medium sized tears, a greater proportion of upper limb surgeons would either consider augmentation or were unsure. Conversely, in large and massive tears the reverse trend was observed, with a greater proportion considering augmentation appropriate in younger age groups (50 or 60 years).

Overall, tear size seemed to be more important than age in assessing patient suitability. Up to half (range 19–57%) would use augmentation in large and massive tears, compared with 10 percent or less (range 3–10%) for small and medium sized rotator cuff tears. However, it is

**Table 3. Types of patch types utilised.**

| Category | n (%) | Count |
|---|---|---|
| *Decellularised Patches* | | |
| *Porcine derived* | 10 (12.6) | |
| Arthrex DX® reinforcement matrix | | 1 |
| Conexa™ reconstructive matrix | | 5 |
| Restore® | | 1 |
| Zimmer® collagen repair patch | | 2 |
| Manufacturer not specified | | 1 |
| *Human derived* | 56 (70.9) | |
| Arthroflex® | | 6 |
| dCell® | | 6 |
| Graftjacket™ | | 44 |
| | | |
| **Type not specified** | 1 (1.3) | 1 |
| **Synthetic Patches** | 12 (15.2) | |
| Artelon® | | 2 |
| Leed-Kuff Patch™ | | 8 |
| Vypro® | | 1 |
| Manufacturer not specified | | 1 |

**Table 4. A summary of motivations behind patch selection.**

| Category | Count (%)* | Total count (n) |
|---|---|---|
| **Product Efficacy** | | 29 |
| Clinical Evidence | 19(24.1) | |
| Personal Experience | 6(7.6) | |
| Peer Experience | 3(3.8) | |
| Regulatory Approval | 1(1.3) | |
| **Product characteristics** | | 18 |
| Material Type | 9(11.4) | |
| Strength | 5(6.3) | |
| Usability | 4(5.1) | |
| **Product Access** | | 17 |
| Cost | 6(7.6) | |
| Local availability | 11(13.9) | |
| **Tear Characteristics** | | 15 |
| Type of Tear | 14(17.7) | |
| Tissue quality | 1(1.3) | |
| **Total responses** | | **61** |
| **Total comments** | | **79** |

*A total of 61 participants provided written responses to the question, *'Why did you use these specific patches'*. Each answer could fit into multiple categories. Percentages are expressed against the total number (79) of extracted themes.

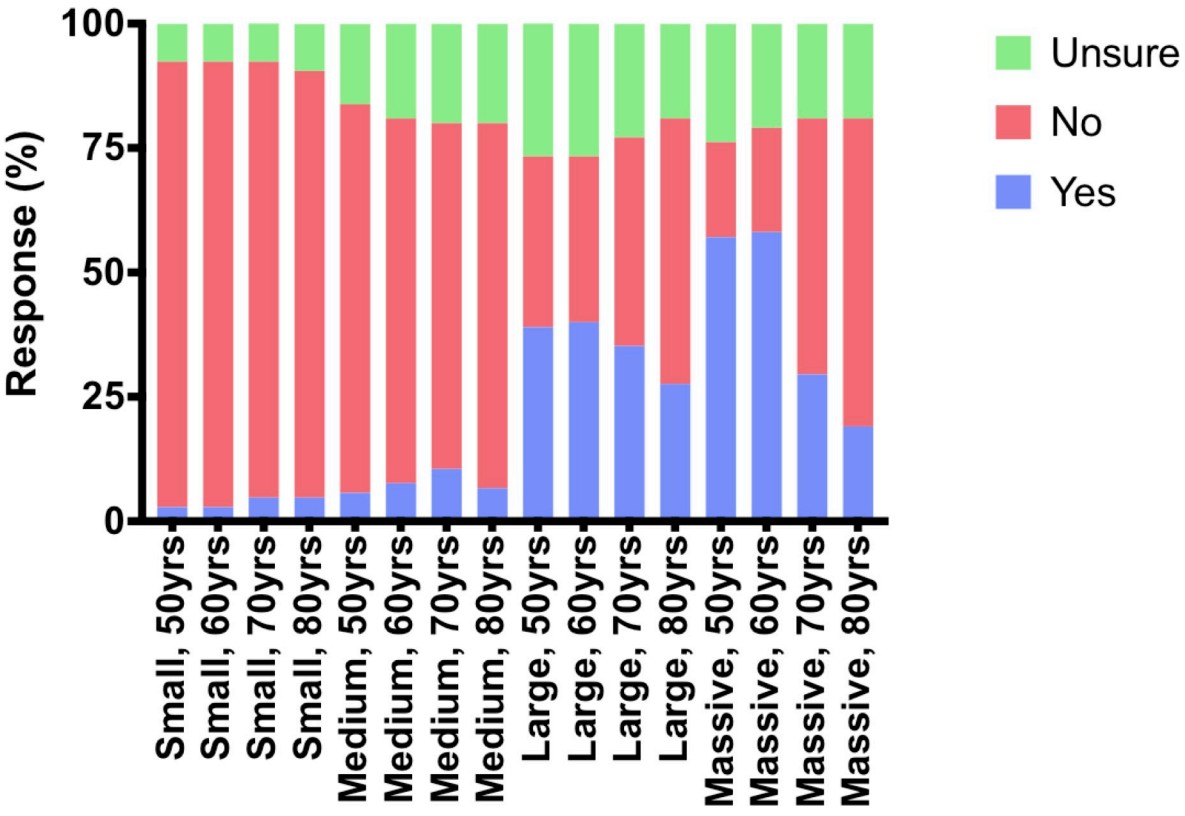

**Fig 1. Perceived suitability of different tear sizes, and age groups, for augmented rotator cuff repair.**

**Table 5. Surgeon reported factors influencing patient selection for patch augmentation.**

| Category | Count (%)* |
|---|---|
| Tear Type | 28(58) |
| Patient Population | 17(35) |
| Previous repair failure | 13(27) |
| Tissue quality | 8(17) |
| No osteoarthritis | 6(13) |
| Supportive evidence or experience | 5(10) |
| **Total responses** | **48** |
| **Total comments** | **77** |

*When asked to give, *'Further comments relating to suitability of a patient for a patch augmented rotator cuff repair'* 48 participants provided free-texts responses. Each answer could fit into multiple categories with 77 comments extracted. Percentages are expressed against the total number (48) of participants responding.

worth noting that considerable uncertainty remains. At least one fifth (range 19–27%) of respondents were unsure as to the role of augmentations in medium, large and massive tears.

48 participants provided additional free text comments, with 6 dominant themes emerging (Table 5). Tear type remained an important consideration during patch augmentation. As well as tear size, the degree of fatty atrophy, intra-operative ability to mobilise the tendon and tension of the repair, were often mentioned as important factors during tear classification. Similarly, a further 17% reported that a surgical assessment of tendon quality was important. A lack of glenohumeral osteoarthritis (12.5%) and failure of a standard repair (27%) were provided as other important qualifiers during augmented repair consideration. Interestingly, for older patients (>70 years) with large tears, the perceived success of reverse shoulder arthroplasty was provided by 10% as justification for avoiding augmented repair.

## Future trial

When asked if they would actively participate in a randomised controlled trial (RCT) of patch augmentation, half of surgeons confirmed an interest with a further 22% undecided and the remainder not interested. 12 respondents provided additional comments that explored barriers to participation. Further trial details were mentioned in almost half (46%) of comments with specific limits placed on the inclusion/exclusion criteria by 23 percent, e.g. *"no compulsion to use patch in small/medium tears"*. A further 31% of comments listed concerns over the type of intervention or comparison that would be utilised. For example, *"compared to balloon interposition"* or *"comparing reverse [total shoulder arthroplasty] with patch repair"*.

## Discussion

Patch augmentation of rotator cuff repair has recently been proposed and its use appears to be increasing. We carried out a survey of surgical society membership to assess current usage and views on it use. This survey has demonstrated a number of insights in to surgeons' opinions around augmentation of rotator cuff repairs. The majority of respondents had used a patch at least once, and tended to use them in larger tears, and with younger patients.

Highly publicised adverse reactions of biomaterials, from metal-on-metal hip replacements through to vaginal mesh implants, has led to increased scrutiny of medical devices and driven regulatory change. Similarly, within the arena of augmented rotator cuff repair, adverse reactions resulted in the market withdrawal of the Restore® patch,[10] and has induced greater

inquiry amongst surgeons. In our survey, this was reflected in surgeons' comments that more data on patch evidence and safety is needed.

Overall, 57% of surgeons had used patches, with 42% of all surgeons using these in the last 6 months. While responders may not be fully representative of the wider surgical community, this survey indicates a substantial uptake of patch augmented repair, a finding that will be clarified in future epidemiological work. The majority of surgeons had used the GraftJacket device, which currently has the highest number of studies published to support its efficacy. This is consistent with the fact that the evidence base and product usability were cited as the most influential factors when choosing a device for augmented repair. Despite this, there were a broad range of other patches (twelve) currently in use, which ranged from decellularised through to synthetic patches. Current reviews of the evidence have shown that, for many of these patches, there is a dearth of robust clinical data for surgeons to base their decisions upon. There remains a clear need for more rigorous evaluation of current and future patch designs. It is our belief that all future trials should combine clinical outcomes with histological analysis, enabling a thorough interrogation of efficacy as well as tissue integration—allowing earlier detection of potential safety concerns.

There was a preference for arthroscopic interventions in those surveyed from University Teaching Hospitals (80%) and DGH's (84%) which is in-keeping with the general trends in greater arthroscopic intervention over time.[4] Surgeons who undertook rotator cuff repair arthroscopically or reported a mixed open/arthroscopic practice were more likely to utilise patch augmentation, which might be reflective of a greater willingness amongst this cohort to adopt new technologies.

The survey also shed light on the relationship between tear size and use of an augmented rotator cuff repair. Patches were more likely to be considered for use in large and massive, rather than small or medium sized tears. The driving force behind this dichotomy remains unclear. It may be that large and massive tears are viewed amongst surgeons as requiring the most supportive healing environment. In fact, small tears in patients aged 80 years are predicted to have a similar chance of repair failure (43%) as massive tears in younger patients (50 years).[5] Given that repair success is intimately linked with symptom resolution,[4] the use of patch augmentation with small to medium tears may gain traction in the future.

As with all survey-based data collection, there is potential for a response bias and the survey frame potentially also limits the generalisability of the findings of this survey. The achieved response rate was low though it was not that dissimilar from that achieved in similar surveys of clinical professional groups and, in our experience, is consistent with other surveys of the BESS surgical membership. BESS members and in particular those who are more likely to respond to a survey like this are not necessarily representative of the wider upper-limb surgical community and may include more research-oriented surgeons. In which case, they may be more familiar with and/or supportive of patch use in rotator cuff repair. Although steps were taken to ensure anonymity, it is possible that respondents may have answered questions in a way that did not reflect their personal beliefs exactly. Few respondents were clearly not in favour of patch use. Moving forward, a wider range of upper-limb surgeons could be consulted to increase the robustness of studies such as this.

Despite the aforementioned limitations, a strong theme from the respondents was the lack of evidence, and robust multi-centre clinical trials. There was appetite to partake in a prospective randomised-control trial, and the need for commonly used patches (e.g. GraftJacket) within the NHS to be included in this RCT was highlighted. A few respondents stated that they were currently involved in or planning a study on patch augmentation.

The general opinion was that more research is required to inform the use (or not) of patch augmentation, with a focus on assessing patient safety and efficacy.

## Conclusions

Amongst UK upper limb surgeons there is a wide range of operator experience with the use of patches to augment rotator cuff repair. There was also substantial uncertainty about which patch to use, and which patient cohorts were most suitable. There is a need for robust clinical evaluation and further research in this area with survey participants expressing substantial support for a future RCT assessing patch augmentation.

## Supporting information

**S1 Survey. Survey instrument.**
(PDF)

**S1 Dataset. Anonymised dataset.**
(XLSX)

## Acknowledgments

We would like to thank all the BESS members who kindly completed this survey.

## Author Contributions

**Conceptualization:** J. L. Rees, A. J. Carr, A. Rangan, M. Thomas, D. J. Beard, J. A. Cook.

**Data curation:** M. J. Baldwin, N. S. Nagra, N. Merritt.

**Formal analysis:** M. J. Baldwin, N. S. Nagra.

**Funding acquisition:** A. J. Carr, D. J. Beard, J. A. Cook.

**Methodology:** M. J. Baldwin, N. S. Nagra, J. L. Rees, A. Rangan, C. Cooper, L. Kottam.

**Project administration:** C. Cooper, L. Kottam.

**Supervision:** J. A. Cook.

**Writing – original draft:** M. J. Baldwin, N. S. Nagra, J. A. Cook.

**Writing – review & editing:** M. J. Baldwin, N. S. Nagra, N. Merritt, J. L. Rees, A. J. Carr, A. Rangan, M. Thomas, D. J. Beard, C. Cooper, L. Kottam, J. A. Cook.

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
