## [Decision Letter · Decision Letter 0]

15 Oct 2019

PONE-D-19-24794

The use of a patch to augment rotator cuff surgery – a survey of UK shoulder and elbow surgeons.

PLOS ONE

Dear Dr Baldwin,

Thank you for submitting your manuscript to PLOS ONE. After careful consideration, we feel that it has merit but does not fully meet PLOS ONE’s publication criteria as it currently stands. Therefore, we invite you to submit a revised version of the manuscript that addresses the points raised during the review process.

You can see that reviewers have divided decision. However, I do think this manuscript has provided some clinical and meaningful information regarding the rotator cuff augmentation, which may help shoulder surgeons to understand current status regarding this treatment. I suggest you fully address the reviewers' concerns including the reasons for the rejection decision that one reviewer has made.

We would appreciate receiving your revised manuscript by Nov 29 2019 11:59PM. To enhance the reproducibility of your results, we recommend that if applicable you deposit your laboratory protocols in protocols.io, where a protocol can be assigned its own identifier (DOI) such that it can be cited independently in the future. For instructions see: http://journals.plos.org/plosone/s/submission-guidelines#loc-laboratory-protocols

We look forward to receiving your revised manuscript.

Kind regards,

Chunfeng Zhao, MD

Academic Editor

PLOS ONE

Journal Requirements:

3. Please provide additional details regarding participant consent. In the ethics statement in the Methods and online submission information, please ensure that you have specified (1) whether consent was suitably informed and (2) what type you obtained (for instance, completion of the questionnaire was deemed to imply consent). If the need for explicit consent was waived by the ethics committee, please include this information.

4. Please note that all PLOS journals ask authors to adhere to our policies for sharing of data and materials: https://journals.plos.org/plosone/s/data-availability. According to PLOS ONE’s Data Availability policy, we require that the minimal dataset underlying results reported in the submission must be made immediately and freely available at the time of publication. As such, please remove any instances of 'unpublished data' or 'data not shown' in your manuscript and replace these with either the relevant data (in the form of additional figures, tables or descriptive text, as appropriate), a citation to where the data can be found, or remove altogether any statements supported by data not presented in the manuscript.

5. Thank you for stating in the manuscript: "This was a voluntary survey of health care professionals therefore formal ethical review was not sought. However, the survey was approved by the BESS committee."

6. We note that you have indicated that data from this study are available upon request. PLOS only allows data to be available upon request if there are legal or ethical restrictions on sharing data publicly. For information on unacceptable data access restrictions, please see http://journals.plos.org/plosone/s/data-availability#loc-unacceptable-data-access-restrictions.

7. Please include your tables as part of your main manuscript and remove the individual files. Please note that supplementary tables (should remain/ be uploaded) as separate "supporting information" files

Reviewers' comments:

Reviewer's Responses to Questions

**Comments to the Author**

1. Is the manuscript technically sound, and do the data support the conclusions?

Reviewer #1: Yes

Reviewer #2: Partly

2. Has the statistical analysis been performed appropriately and rigorously? 

Reviewer #1: Yes

Reviewer #2: No

3. Have the authors made all data underlying the findings in their manuscript fully available?

Reviewer #1: Yes

Reviewer #2: No

4. Is the manuscript presented in an intelligible fashion and written in standard English?

Reviewer #1: Yes

Reviewer #2: Yes

5. Review Comments to the Author

Reviewer #1: Patch has been increasingly used in rotator cuff repair, especially for large/massive rotator cuff tear. But there are limited epidemiologic studies about the choice of patch. This paper focused to survey the current UK practice and opinion relating to the factors that influence choice of path, current patient selection and willingness to assist with generation of improved evidence. This study was well performed and provided the panorama of patch uses in the UK. Here are some questions for the authors.

1. Why the authors choose the 6 months as one time point in the survey contents?

2. The date was collected from different surgeons. So please clarify how to validate these date.

3. Why did you preclude the ages less than 50 in the survey contents?

4. Arthroscopic rotator cuff repairs accounted for 66%, but this was not according with the data in Table 2.

5. Clarify the meaning of “Number of patches implanted” in Table 2. Does that mean the number of patch used in one surgery or the total number of patch used by one surgeon?

Reviewer #2: Dear Authors,

Thank you for submitting this manuscript.

In this paper, you presented a survey of UK shoulder and elbow surgeons, summarizing the patches which have been used in rotator cuff repair surgeries.

This paper is well written and well thought out. However, there is no evidence-based data to contribute to clinical practice. Also, the conclusion of this study is irrelevant and inconclusive.

Therefore, this manuscript is not suitable for publication.

6. PLOS authors have the option to publish the peer review history of their article (what does this mean?). If published, this will include your full peer review and any attached files.

Reviewer #1: No

Reviewer #2: No

---

## [Author Response · Author response to Decision Letter 0]

28 Nov 2019

Following consideration of the reviewer’s comments, appropriate changes have now been incorporated into the article and are summarised below. 

Referee 1 Comments to Author: 

1. Why the authors choose the 6 months as one time point in the survey contents? 

Respondents were asked if they had used a patch to augment the surgical repair of the rotator cuff within the last 6 months. This question was designed to understand current patch usage rather than total career patch use. Given recent concerns over patch safety and an increasingly large array of patches available we felt this was an important consideration. During survey design we felt that a short time period would minimise recall bias, but that this also needed to be of long enough during to capture ‘events’ i.e. the use of rotator cuff repair. A six-month duration, whilst arbitrary, was felt by the authors to best balance these considerations. 

2. The date was collected from different surgeons. So please clarify how to validate these date.

Data collected from each surgeon was not validated. Whilst this may have been possible, for example through access to the operative log-book for each surgeon, these records are themselves often incomplete. It is also likely that any such strategy would have significantly impacted upon survey uptake. However, we have added the following sentence to acknowledge this methodological limitation; ‘No attempt was made to validate individual responses’

3. Why did you preclude the ages less than 50 in the survey contents?

Previous work has shown that over 90% of rotator cuff tears occur in the over 50s (Yamamoto et al. J Shoulder Elbow Surg (2010) 19, 116-120). Those under the age of 50 are more likely to be of a different aetiology e.g. traumatic tears. We sought to characterise patch usage for the ‘general’ rotator cuff tear and not for specific clinical indications e.g. trauamatic vs degenerative, superior cuff vs anterior tendon tears. 

4. Arthroscopic rotator cuff repairs accounted for 66%, but this was not according with the data in Table 2. 

Many thanks for pointing out this important typographical error in Table 2. This has now been corrected.

5. Clarify the meaning of “Number of patches implanted” in Table 2. Does that mean the number of patch used in one surgery or the total number of patch used by one surgeon?

The number of patches implanted refers to the total number of patches used by each respondent during their career to date. The table legend has now been amended to reflect this: “n refers to the number of respondents. Total number of patches refers to the total number of patches used during each respondents surgical career to date.”

Reviewer 2 Questions:

It is difficult to response to the second reviewers’ concerns, as few reasons for their recommended rejection of the paper were provided. We would however raise that while the reviewer has cited statistical concerns, that the final author is an experienced academic statistician. We would thank the reviewer for their comment that “This paper is well written and well thought out”. However, would disagree that “there is no evidence-based data to contribute to clinical practice”. For the first time we have provided data on patch usage and patch choice within a public healthcare system. Understanding the clinical practice of our colleagues is directly relevant to current clinical practice, and vital in the planning of future clinical trials that seek to comprehensively assess and evidence our clinical and surgical decisions during rotator cuff repair. 

Academic Editors points:

1. PLOS style requirement – The manuscript has been re-formatted to meet the requirements of the various style templates (main body/tables and authors affiliations) 

2. Include captions for supporting information – A caption for the supplementary files has been provided

3. Please provide additional details regarding participant consent – The following sentence has been added into the methods section: “This was a voluntary survey of health care professionals therefore formal ethical review was not sought. However, the survey was approved by the BESS committee. A formal consenting process was not undertaken, rather completion of the survey was taken as implied consent.”

4. Please remove any instances of 'unpublished data' or 'data not shown' in your manuscript – Completed as requested 

5. Thank you for stating in the manuscript: "This was a voluntary survey of health care professionals therefore formal ethical review was not sought. However, the survey was approved by the BESS committee." Please amend your current ethics statement to include the full name of the ethics committee/institutional review board(s) that approved your specific study. – Changed from ‘BESS’ to ‘British Elbow and Shoulder Society’

6. We note that you have indicated that data from this study are available upon request - Anonymised dataset has been provided as a supplementary file (S2_Dataset)

7. Please include your tables as part of your main manuscript and remove the individual files – Completed as requested

---

## [Decision Letter · Decision Letter 1]

26 Feb 2020

The use of a patch to augment rotator cuff surgery – a survey of UK shoulder and elbow surgeons.

PONE-D-19-24794R1

Dear Dr. Baldwin,

We are pleased to inform you that your manuscript has been judged scientifically suitable for publication and will be formally accepted for publication once it complies with all outstanding technical requirements.

With kind regards,

Chunfeng Zhao, MD

Academic Editor

PLOS ONE

Additional Editor Comments (optional):

Reviewers' comments:

Reviewer's Responses to Questions

**Comments to the Author**

1. If the authors have adequately addressed your comments raised in a previous round of review and you feel that this manuscript is now acceptable for publication, you may indicate that here to bypass the “Comments to the Author” section, enter your conflict of interest statement in the “Confidential to Editor” section, and submit your "Accept" recommendation.

Reviewer #2: All comments have been addressed

2. Is the manuscript technically sound, and do the data support the conclusions?

Reviewer #2: Yes

3. Has the statistical analysis been performed appropriately and rigorously? 

Reviewer #2: Yes

4. Have the authors made all data underlying the findings in their manuscript fully available?

Reviewer #2: Yes

5. Is the manuscript presented in an intelligible fashion and written in standard English?

Reviewer #2: Yes

6. Review Comments to the Author

Reviewer #2: (No Response)

7. PLOS authors have the option to publish the peer review history of their article (what does this mean?). If published, this will include your full peer review and any attached files.

Reviewer #2: No

---

## [Editor Report · Acceptance letter]

5 Mar 2020

PONE-D-19-24794R1 

The use of a patch to augment rotator cuff surgery – a survey of UK shoulder and elbow surgeons. 

Dear Dr. Baldwin:

I am pleased to inform you that your manuscript has been deemed suitable for publication in PLOS ONE. Congratulations! Your manuscript is now with our production department. 

With kind regards,

on behalf of

Dr. Chunfeng Zhao 

Academic Editor

PLOS ONE